# Association between nutritional status and dengue severity in Thai children and adolescents

**Haypheng Te**[1,2], **Pimolpachr Sriburin**[1], **Jittraporn Rattanamahaphoom**[1], **Pichamon Sittikul**[1], **Weerawan Hattasingh**[1], **Supawat Chatchen**[1], **Salin Sirinam**[1], **Kriengsak Limkittikul**[1]*

1 Department of Tropical Pediatrics, Faculty of Tropical Medicine, Mahidol University, Bangkok, Thailand,
2 Division of Infectious Diseases, National Pediatric Hospital, Ministry of Health, Phnom Penh, Cambodia

* kriengsak.lim@mahidol.ac.th

**Data Availability Statement:** Data are available from Mendeley Data, doi: 10.17632/mv86fths7g.1 (https://data.mendeley.com/datasets/mv86fths7g/).

## Abstract

Most cases of dengue virus infection are mild, but severe cases can be fatal. Therefore, identification of factors associated with dengue severity is essential to improve patient outcomes and reduce mortality. The objective of this study was to assess associations between nutritional status and dengue severity among Thai children and adolescents. This retrospective cross-sectional study was based on the medical records of 355 patients with dengue treated at the Hospital for Tropical Disease (Bangkok, Thailand) from 2017 to 2019. Subjects were Thai children aged less than 18 years with dengue virus infection confirmed by positive NS1 antigen or IgM. The 1997 and 2009 World Health Organization (WHO) dengue classifications were used to define disease severity and body mass index for age while the WHO growth chart was used to classify nutritional status. The proportions of patients with dengue fever who were underweight, normal weight, and overweight were 8.8%, 61.5%, and 29.7%, respectively. The proportions of patients with dengue haemorrhagic fever (DHF) who were underweight, normal weight, and overweight were 10.2%, 66.1%, and 23.7%, respectively. The proportions of patients with non-severe dengue who were underweight, normal weight, and overweight were 8.6%, 60.9%, and 30.5%, respectively; the same proportions of patients with severe dengue were 10.5%, 67.1%, and 22.4%, respectively. Higher proportions of patients with severe plasma leakage (DHF grade III and IV) were overweight compared with those with mild plasma leakage (DHF grade I and II) (45.5% vs. 18.8%). No difference in nutritional status was observed in patients with different dengue severity.

## Author summary

Dengue is a rapid spreading mosquito-borne viral infection. Infections cause mild to severe diseases, including dengue haemorrhagic fever (DHF), a severe form that may kill infants and young children. One potential factor associated to dengue severity is the patients' nutritional status. For dengue severity, World Health Organization (WHO)

**Funding:** The authors received no specific funding for this work.

**Competing interests:** The authors have declared that no competing interests exist.

dengue case classification criteria in 1997 and revised version in 2009 were used. While body mass index for age and WHO growth chart were used to classify nutritional status. In this study, we would like to demonstrate the association of underweight/overweight and severity of dengue infection using both WHO classifications in 355 Thai children and adolescents. We compared the proportions of patients with dengue fever with different nutritional status to those with dengue haemorrhagic fever (DHF), and also compared non-severe dengue to severe dengue. Nevertheless, we could not demonstrate an association of nutritional status and dengue severity. It was noted that higher proportions of patients with severe plasma leakage (DHF grade III and IV) were overweight compared to those with mild plasma leakage (DHF grade I and II) (45.5% vs. 18.8%).

## Introduction

Dengue is a common mosquito-borne disease and is distributed worldwide, especially in tropical and subtropical areas. The disease results from infection by dengue virus, a positive single-stranded RNA virus in the family *Flaviviridae* [1]. More than a hundred countries have recorded cases of dengue. In 2010, it was estimated that the numbers of detected and undetected dengue infections worldwide were 294 million and 96 million, respectively [2].

Most individuals with dengue will experience mild illness, but severe cases and deaths represent one third and two percent of those who are hospitalized, respectively [3]. Therefore, dengue infection remains a global health concern, and recognition of severe disease is necessary to decrease mortality and improve patient outcomes. However, identification of factors associated with severe dengue is challenging, and the roles of some factors and their impacts on dengue infection remain unclear. Moreover, nutritional status is also a global health issue including both under-nutrition and over-nutrition. According to a malnutrition report from the World Health Organization (WHO), more than 200 million children were stunted, wasting, or overweight; additionally, 45% of deaths among children under the age of 5 years were associated with nutritional factors [4]. Both under and over nutrition are health issue. In Thailand, overweight had been appeared an emerging disorder. The prevalence of obesity from 3 to 18 years old of children and adult were 9.1% and 6.5% respectively [5]. However, the rate could be changed from urban and rural population.

Significantly, nutrition was matter to immune functions which it could influence on genomics and metabolisms, and the role of adipose tissue in overweight could stimulate more inflammatory mediators which leading to increase capillary permeability and plasma leakage [6,7]. However, whether nutritional status is really the risk of severe dengue, there might need more further evidences. The potential association between nutritional status and dengue severity remains controversial. A systematic review conducted in 2016 was unable to show a significant association between nutritional status and dengue severity [8]. Therefore, the objective of this study was to identify associations between nutritional status and dengue severity in children and adolescents. These data will contribute to the evidence base regarding the role of nutritional status in dengue infection and may raise awareness among clinicians and researchers to improve patient's outcome.

## Materials and methods

### Ethics statement

This study was approved by the director of the Hospital for Tropical Diseases and Ethics Committee of the Faculty of Tropical Medicine, Mahidol University (EC Approval MUTM 2020-

020-01). The formal consent was not obtained due to the retrospective study design. However, all medical records were anonymous.

## Study design and setting

This was a cross-sectional study conducted at the Hospital for Tropical Diseases, Bangkok, Thailand. All available medical records of paediatric patients who diagnosed dengue infection were retrieved from both outpatient and inpatient departments from 1 January 2017 to 31 December 2019. All Thai children aged less than 18 years with acute dengue infection confirmed by positive NS1 antigen or IgM were included. Patients were excluded if data were unavailable on nutritional status (e.g. age, weight and height) and/or dengue severity (e.g. haematocrit on defervescent date and its baseline, vital signs and bleeding information). Pre-set Case Record Form were used to check eligibilities and extracted specific information related to dengue severity and nutritional status from patient records.

## Operating definitions

For assessing nutritional status, patient weight and height were measured on the first day of hospitalization and used to calculate body mass index (BMI). The 2006 WHO growth chart for children aged 0 to 5 years [9] and the 2007 WHO growth chart for children aged 5 to 19 years were used to classify nutritional status [10]. In accordance with the 2006 WHO growth chart, underweight was defined as < -2 standard deviations (SD), normal weight was defined as -2 SD to + 2 SD, overweight was defined as > +2 SD, and obesity was defined as > +3 SD. In accordance with the 2007 WHO growth chart, underweight was defined as < -2 SD, normal weight was defined as -2 SD to +1 SD, overweight was defined as > +1 SD, and obesity was defined as > +2 SD.

The 1997 and 2009 WHO guidelines on dengue infection were used to classify dengue severity. Diagnoses of dengue fever (DF) and dengue haemorrhage fever (DHF) were made after the critical phase in accordance with the 1997 WHO classification, and diagnoses of severe dengue were made in accordance with the 2009 WHO classification [11,12]. The day of defervescence was defined as when fever had resolved for at least 24 hours. For in case of dengue classification disagreement between final diagnosis in medical record and investigator's judgment, it would be re-justified by the group of independent paediatricians.

## Statistical analysis

SPSS version 18.0 (SPSS Inc., Chicago, IL, USA) was used for statistical analysis. Data were summarized using descriptive statistical methods including frequencies, percentages, proportions, median, and interquartile ranges (IQR). The Chi-square test as well as odd ratios (ORs) and 95% confidence intervals (CIs) were used to assess associations between categorical variables. Non-parametric tests were used to determine the difference between continuous variables. Two-tailed values of $P<0.05$ were considered statistically significant.

## Results

### Demographic data

A total of 574 medical records were screened against the inclusion and exclusion criteria. After review, 355 cases were included and 219 cases were excluded as age were greater than 17 years (112 cases), non-Thai (37 cases), unavailable result of laboratory confirmed dengue infection (44 cases), and unavailable clinical data (26 cases). Among these 355 patients, 243 (68.4%) were inpatients and 179 (50.4%) visited the hospital in 2018. Most patients (79.4%) were

between the ages of 11 and 17 years with a median age of 15 years. The male to female ratio was 1.6:1. Fifty-two (14.6%) patients had underlying diseases, most commonly allergic rhinitis (7.9%), G6PD deficiency (2.3%), and thalassemia minor (1.7%). Seventeen (4.8%) patients had co-infections. Most patients (99.7%) fully recovered. The one patient who died was an obese 8-year-old boy who presented with profound shock, severe plasma leakage, and bleeding with fluid overload (Table 1).

## Association between nutritional status and dengue severity

Tables 2 and 3 show the distribution of dengue severity according to the 1997 and 2009 WHO classifications and nutritional status. Twenty-six (8.8%), 182 (61.5%), and 88 (29.7%) patients with DF (n = 296) were underweight, normal weight, and overweight, respectively. Six (10.2%), 39 (66.1%), and 14 (23.7%) patients with DHF (n = 59) were underweight, normal weight, and overweight, respectively. According to the WHO 2009 classification, 24 (8.6%), 170 (60.9%), and 85 (30.5%) patients with non-severe dengue (n = 279) were underweight, normal weight, and overweight, respectively while 8 (10.5%), 51 (67.1%), and 17 (22.4%) patients with severe dengue (n = 76) were underweight, normal weight, and overweight, respectively.

Tables 4 and 5 show associations between nutritional status and dengue severity. The odds ratio (OR) of underweight in patients with DHF was 1.17 (95% CI 0.46–2.99, P = 0.73) while that of overweight was 0.73 (95% CI 0.38–1.40, P = 0.35). The OR of underweight in patients with severe dengue was 1.25 (95% CI 0.53–2.90, P = 0.60) while that of overweight was 0.65 (95% CI 0.36–1.19, P = 0.16). Thus, there were no statistically significant associations between nutritional status and DHF or severe dengue. However, the proportion of patients with dengue shock syndrome (DHF grade III and IV) who were overweight (5/11; 45.5%) was higher than the same proportion among patients with mild plasma leakage (DHF grade I and II) (9/48; 18.8%) (OR 3.61, 95% CI 0.90–14.5, P = 0.06) as shown in Table 2.

## Discussion

The overall distribution of nutritional status among children with dengue in this study was as follows: underweight (9.0%), normal weight (62.3%), and overweight/obese (28.7%). This finding suggested that a relatively high proportion of children with dengue are overweight in Thailand. However, our results were derived from a single study centre located in the capital city. The study site was also a tertiary care hospital and therefore the characteristics of patients tended to be more serious. In this study, the proportions of patients with DHF (16.6%) and severe dengue (21.4%) may not reflect the general distribution of dengue severity in the Thai population. The result of study also demonstrated higher proportion of severe dengue and DHF in the older children which may relate to more severity in the secondary dengue infection. In this study, 52 (14.6%) patients who had underlying diseases were included in analysis because they were mild conditions and had no difference in distribution among dengue severity.

The major goal of this study was to assess associations between nutritional status and dengue severity. as human immunity was lined with nutrition. Weight for height was used to classify nutritional status, weight on the first day of visit may not reflect the ideal underlying weight due to loss of appetite, however it was the practical underlying weight. In addition, BMI for age of WHO growth charts were used as they could be accepted widely and internationally by others. Similar patterns of distribution of nutritional status were observed in patients with DF vs. DHF and patients with non-severe vs. severe dengue: most patients had normal weights, followed by overweight and underweight. This finding may vary depending

**Table 1. Demographic data, nutritional status and dengue clinical classifications.**

| Characteristics | Total (n = 355) | OPD cases (n = 112) | IPD cases (n = 243) | P-value |
|---|---|---|---|---|
| Age, years [median (IQR)] | 15 (12–16) | 13 (9–16) | 15 (13–16) | 0.001 |
| Age groups, n (%) | | | | |
| • 0–2 years | 2 (0.6) | 1 (0.9) | 1 (0.4) | 0.57 |
| • 3–5 years | 15 (4.2) | 8 (7.1) | 7 (2.9) | 0.64 |
| • 6–10 years | 56 (15.8) | 28 (25.0) | 28 (11.5) | 0.01 |
| • 11–14 years | 103 (29.0) | 32 (28.6) | 71 (29.2) | 0.9 |
| • 15–17 years | 179 (50.4) | 43 (38.4) | 136 (56.0) | 0.02 |
| Male: female, n (ratio) | 221: 134 (1.6: 1) | 63: 49 (1.2: 1) | 158: 85 (1.8: 1) | 0.11 |
| Year, n (%) | | | | |
| • Year 2017 | 77 (21.7) | 24 (21.4) | 53 (21.8) | 0.57 |
| • Year 2018 | 179 (50.4) | 52 (46.4) | 127 (52.3) | 0.30 |
| • Year 2019 | 99 (27.9) | 36 (32.1) | 63 (25.9) | 0.22 |
| Length of hospitalization, days [median (IQR)] | 4 (3–5) | n/a | 4 (3–5) | n/a |
| Prior medical consultation, n (%) | 204 (57.5) | 34 (30.4) | 170 (70.0) | < 0.01 |
| Duration of fever at first visit, n (%) | | | | |
| • 1–3 days | 241 (67.9) | 86 (76.8) | 155 (63.8) | 0.01 |
| • ≥ 4 days | 114 (32.1) | 26 (23.2) | 88 (36.2) | |
| Underlying diseases, n (%) | | | | |
| • Allergy rhinitis | 28 (7.9) | 12 (10.7) | 16 (6.6) | 0.18 |
| • G6PD deficiency | 8 (2.3) | 0 | 8 (3.3) | 0.05 |
| • Thalassemia minor | 6 (1.7) | 0 | 6 (2.4) | 0.09 |
| • Others | 10 (2.8) | 0 | 10 (4.1) | 0.02 |
| Co-infection, n (%) | | | | |
| • Bacterial infection | 6 (1.7) | 0 | 6 (2.4) | 0.09 |
| • Viral infection | 4 (1.1) | 3 (2.6) | 1 (0.4) | 0.06 |
| • Sepsis (unknown cause) | 7 (2.0) | 0 | 7 (2.9) | 0.07 |
| Outcome, n (%) | | | | |
| • Recovery | 354 (99.7) | 112 (100) | 242 (99.6) | 0.49 |
| • Death | 1 (0.3) | 0 | 1 (0.4) | |
| Nutritional status, n (%) | | | | |
| • Underweight | 32 (9.0) | 10 (8.9) | 22 (9.1) | 0.97 |
| • Normal | 221 (62.3) | 70 (62.5) | 151 (62.1) | 0.94 |
| • Overweight* | 102 (28.7) | 32 (28.6) | 70 (28.8) | 0.96 |
| (Obesity) | 57 (16.0) | 17 (15.2) | 40 (16.5) | 0.76 |
| Diagnosis according to WHO 1997 classification, n (%) | | | | |
| • DF | 296 (83.4) | 109 (97.3) | 187 (77.0) | <0.01 |
| • DHF grade I | 36 (10.1) | 3 (2.7) | 33 (13.6) | 0.02 |
| • DHF grade II | 12 (3.4) | 0 | 12 (4.9) | 0.01 |
| • DHF grade III | 10 (2.8) | 0 | 10 (4.1) | 0.02 |
| • DHF grade IV | 1 (0.3) | 0 | 1 (0.4) | 0.49 |
| Diagnosis according to WHO 2009 classification, n (%) | | | | |
| • Non-severe dengue | 279 (78.6) | 109 (97.3) | 170 (70.0) | <0.01 |
| • Severe dengue | 76 (21.4) | 3 (2.7) | 73 (30.0) | |

Abbreviations: IQR, interquartile range; OPD, out-patient department; IPD, In-patient department; WHO, World Health Organization; DF, dengue fever; DHF, dengue haemorrhagic fever; G6PD, glucose-6-phosphate dehydrogenase n/a: Not applicable

*Overweight: > +2 SD BMI for ages 0 to <5 years; > +1 SD BMI for ages 5 to 17 years.

**Table 2. Patient's characteristics by 1997 WHO classification.**

| Patient's characteristics [n(%)] | DF (n = 296) | DHF (n = 59) | | | | P-value DF vs DHF | P-value DHF gr. I, II vs DHF gr. III, IV |
|---|---|---|---|---|---|---|---|
| | | gr. I (n = 36) | gr. II (n = 12) | gr. III (n = 10) | gr. IV (n = 1) | | |
| **Age 15–17 years** | 141 (47.6) | 24 (66.7) | 6 (50.0) | 8 (80.0) | 0 | 0.01 | 0.52 |
| **Male** | 180 (60.8) | 24 (66.7) | 9 (75.0) | 7 (70.0) | 1 (100) | 0.20 | 0.79 |
| **IPD** | 187 (63.2) | 33 (91.7) | 12 (100) | 10 (100) | 1 (100) | <0.01 | 0.39 |
| **Underlying diseases** | 44 (14.9) | 6 (16.7) | 2 (16.7) | 1 (10.0) | 0 | 0.79 | 0.63 |
| **Clinical presentation** | | | | | | | |
| Clinical bleeding | 50 (18.9) | 0 | 12 (100) | 0 | 1 (100) | 0.34 | 0.25 |
| Major organ involvement | | | | | | | |
| • Transaminitis (ALT >100U/L) | 40/196 (20.4) | 10/33 (30.3) | 3 (25.0) | 3 (30.0) | 1 (100) | 0.11 | 0.62 |
| • Creatinine > 1.5 mg/dL | 0 | 0 | 0 | 0 | 1 (100) | 0.09 | 0.11 |
| • Respiratory distress | 2 (0.67) | 0 | 0 | 2 (20.0) | 1 (100) | <0.01 | <0.01 |
| **Treatments** | | | | | | | |
| • Colloidal solution | 0 | 1 (2.7) | 1 (8.3) | 3 (30.0) | 1 (100) | <0.01 | 0.01 |
| • Blood transfusion | 0 | 0 | 2 (16.7) | 2 (20.0) | 1 (100) | <0.01 | 0.01 |
| **Nutritional status** | | | | | | | |
| • Underweight | 26 (8.8) | 4 (11.2) | 2 (16.7) | 0 | 0 | 0.73 | 0.58 |
| • Normal | 152 (61.5) | 25 (69.4) | 8 (66.7) | 6 (60.0) | 0 | 0.50 | 0.37 |
| • Overweight | 88 (29.7) | 7 (19.4) | 2 (16.7) | 4 (40.0) | 1 (100) | 0.35 | 0.06 |
| (Obesity) | 50 (16.9) | 3 (8.3) | 2 (16.7 | 1 (10.0) | 1 (100) | 0.33 | 0.60 |

Abbreviations: WHO, World Health Organization; DF, dengue fever; DHF, dengue haemorrhage fever; gr.,grade; n, number; IPD, in-patient department; ALT, alanine transaminase; U/L, units per litre; mg/dL, milligrams per litre

on the criteria used to classify nutritional status and dengue severity. We did not identify any statistically significant associations between dengue severity and nutritional status, although higher proportions of patients with dengue shock syndrome (DHF grade III/IV) (5/11; 45.5%) were overweight/obese compared with those with DHF grades I and II (9/48; 18.8%). However, this result may have arisen from the small number of patients with severe dengue studied.

Using another type of statistical analysis, Z-scores for the BMI in each patient which may provide a fine details of nutrition status, were also calculated using the WHO growth charts as references. Independent t-tests comparing Z-scores in patients with different dengue severity did not reveal any associations between dengue severity and nutritional status (S1 and S2 Tables).

In a previous study applying the 1997 WHO criteria, the distribution of nutritional status among a large group of Thai children with DF and DHF was similar to that observed in our study: the majority of children were normal weight, followed by overweight and then underweight [13]. However, the study used weight for age and a Thai growth chart to classify nutritional status. Another study applied the 2009 WHO criteria and found a slightly different distribution of nutritional status compared with our study: the proportions of underweight and overweight in patients with non-severe dengue or severe dengue without shock were 14% and 23%, respectively, while in patients with dengue shock syndrome these proportions were 11.6% and 27%, respectively [14]. In agreement with our findings, a systematic review found that the ORs (95% CIs) of dengue shock syndrome compared with DHF grade I/II in overweight and underweight children were 1.31 (0.91–1.88) and 1.17 (0.99–1.39), respectively [8].

**Table 3. Patient's characteristics by 2009 WHO classification.**

| Patient's characteristics [n(%)] | Non-severe dengue (n = 279) | Severe dengue (n = 76) | | | | P-value |
| --- | --- | --- | --- | --- | --- | --- |
| | | Total (n = 76) | Severe bleeding (n = 29) | Plasma leakage / shock (n = 59) | Major organ involvement (n = 24) | non-severe vs severe |
| Age 15–17 years | 133 (37.7) | 46 (60.5) | 13 (44.8) | 38 (64.4) | 18 (75.0) | 0.04 |
| Male | 179 (64.2) | 42 (55.3) | 10 (34.5) | 41 (69.5) | 18 (75.0) | 0.15 |
| IPD | 170 (60.9) | 73 (96.1) | 29 (100) | 56 (94.9) | 24 (100) | <0.01 |
| Underlying diseases | 43 (15.4) | 9 (11.8) | 2 (6.9) | 8 (13.6) | 3 (12.5) | 0.43 |
| Major organ involvement | | | | | | |
| • Transaminitis (ALT >100U/L) | 35/182 (19.2) | 22/70 (31.4) | 8/26 (30.8) | 17/56 (30.4) | 22 (91.7) | 0.03 |
| • Creatinine > 1.5 mg/dL | 0 | 1/38 (2.6) | 1/12 (8.3) | 1/34 (2.9) | 1/15 (6.7) | 0.12 |
| • Respiratory distress | 2 (0.7) | 3 (3.9) | 1 (3.4) | 3 (5.1) | 3 (12.5) | 0.03 |
| Treatment | | | | | | |
| • Colloidal solution | 0 | 5 (6.6) | 1 (3.4) | 5 (8.5) | 2 (8.3) | <0.01 |
| • Blood transfusion | 0 | 5 (6.6) | 3 (10.3) | 5 (8.5) | 3 (12.5) | <0.01 |
| Nutritional status | | | | | | |
| • Underweight | 24 (8.6) | 8 (10.5) | 4 (13.8) | 6 (10.2) | 1 (4.2) | 0.60 |
| • Normal | 170 (60.9) | 51 (67.1) | 19 (65.5) | 39 (66.1) | 18 (75) | 0.32 |
| • Overweight | 85 (30.5) | 17 (22.4) | 6 (20.7) | 14 (23.7) | 5 (20.8) | 0.16 |
| (Obesity) | 49 (17.6) | 8 (10.5) | 4 (13.8) | 7 (11.9) | 3 (12.5) | 0.13 |

Abbreviations: WHO, World Health Organization; n, number; IPD, in-patient department; ALT, alanine transaminase; U/L, units per litre; mg/dL, milligrams per litre

**Table 4. Association between nutritional status and dengue severity according to the 1997 WHO classification.**

| Nutritional status [n(%)] | DHF (n = 59) | DF (n = 296) | odds ratio (95% CI) | P-value |
| --- | --- | --- | --- | --- |
| Underweight (n = 32) | 6 (10.2) | 26 (8.8) | 1.17 (0.46.-2.99) | 0.73 |
| Overweight (n = 102) | 14 (23.7) | 88 (29.7) | 0.73 (0.38–1.40) | 0.35 |
| Obese (n = 57) | 7 (11.9) | 50 (16.9) | 0.66 (0.28–1.54) | 0.33 |

Abbreviations: DHF, dengue haemorrhagic fever; DF, dengue fever; CI, confidential interval

**Table 5. Association between nutritional status and dengue severity according to the 2009 WHO classification.**

| Nutritional status [n(%)] | Severe dengue (n = 76) | Non-severe dengue (n = 279) | odds ratio (95% CI) | P-value |
| --- | --- | --- | --- | --- |
| Underweight (n = 32) | 8 (10.5) | 24 (8.6) | 1.25 (0.53–2.90) | 0.60 |
| Overweight (n = 102) | 17 (22.4) | 85 (30.5) | 0.65 (0.36–1.19) | 0.16 |
| Obese (n = 57) | 8 (10.5) | 49 (17.6) | 0.55 (0.24–1.22) | 0.13 |

Abbreviation: CI, confidential interval

The systematic review also showed that 25% of obese children developed severe dengue compared with 22.7% of non-obese children (OR 1.38; 95% CI 1.10–1.73) [15]. However, these results were derived from pooled data from 15 studies that used various classifications of nutritional status.

## Conclusions

The distributions of nutritional status by dengue severity using the 1997 and 2009 WHO classifications were similar. However, there was a trend toward higher prevalence of overweight in patients with dengue shock syndrome compared with those with mild plasma leakage (45.5% vs. 18.8%). Larger prospective studies may be necessary to more accurately assess associations between nutritional status and dengue severity.

## Supporting information

**S1 Table. Z score of BMI and dengue severity according to the 1997 WHO classification.** (DOCX)

**S2 Table. Z score of BMI and dengue severity according to the 2009 WHO classification.** (DOCX)

## Acknowledgments

We thank the Hospital for Tropical Disease of Mahidol University for approving the study and allowing us to collect these data. We thank Edanz (https://www.edanz.com/ac) for editing a draft of this manuscript.

## Author Contributions

**Conceptualization:** Haypheng Te, Weerawan Hattasingh, Supawat Chatchen, Kriengsak Limkittikul.

**Data curation:** Haypheng Te, Pimolpachr Sriburin, Jittraporn Rattanamahaphoom, Pichamon Sittikul.

**Formal analysis:** Haypheng Te, Kriengsak Limkittikul.

**Investigation:** Haypheng Te, Pimolpachr Sriburin, Jittraporn Rattanamahaphoom, Pichamon Sittikul, Weerawan Hattasingh, Supawat Chatchen, Salin Sirinam, Kriengsak Limkittikul.

**Methodology:** Haypheng Te, Pimolpachr Sriburin, Jittraporn Rattanamahaphoom, Salin Sirinam, Kriengsak Limkittikul.

**Project administration:** Kriengsak Limkittikul.

**Supervision:** Weerawan Hattasingh, Supawat Chatchen, Salin Sirinam, Kriengsak Limkittikul.

**Writing – original draft:** Haypheng Te.

**Writing – review & editing:** Kriengsak Limkittikul.

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
