## [Decision Letter · Decision Letter 0]

22 Nov 2021

Dear Dr. Limkittikul,

Thank you very much for submitting your manuscript "Association between nutritional status and dengue severity in Thai children" for consideration at PLOS Neglected Tropical Diseases. As with all papers reviewed by the journal, your manuscript was reviewed by members of the editorial board and by several independent reviewers. In light of the reviews (below this email), we would like to invite the resubmission of a significantly-revised version that takes into account the reviewers' comments. 

We cannot make any decision about publication until we have seen the revised manuscript and your response to the reviewers' comments. Your revised manuscript is also likely to be sent to reviewers for further evaluation.

Sincerely,

Olaf Horstick, FFPH(UK)

Associate Editor

Sergio Recuenco

Deputy Editor

Reviewer's Responses to Questions

**Key Review Criteria Required for Acceptance?**

**Methods**

-Are the objectives of the study clearly articulated with a clear testable hypothesis stated?

-Is the study design appropriate to address the stated objectives?

-Is the population clearly described and appropriate for the hypothesis being tested?

-Is the sample size sufficient to ensure adequate power to address the hypothesis being tested?

-Were correct statistical analysis used to support conclusions?

-Are there concerns about ethical or regulatory requirements being met?

Reviewer #1: The study design is not fully appropriate for the stated objectives.

• Suggest to compare OPD and IPD group to see any different in their nutritional status or not and analyze each group separately?

• Suggest to add more detail of clinical and hospital management, eg. complications (hepatitis, AKI, bleeding,…), Dextran, blood transfusion, Length of stay in the hospital

Reviewer #2: The authors did not describe a designed retrospective chart review with the classification of dengue severity based on both the 1997 and 2009 Dengue WHO guidelines. 

- They did not show how they sample, record, and review the dengue cases as well as confirm the classifications.

- Furthermore, they should clarify why they included the dengue cases with underlying diseases in the study.

Reviewer #3: o 0-17 years is a *huge* age range. I would strongly suggest to do some sort of adjustment or stratification across age groups. It is possible that if an association exists, it is different in the very young vs. adolescents. 

o I would also suggest adjusting for sex of the child in addition.

o Is there a reason why 11-17y are together in one group? This is almost 80% of the sample, could split this group.

**Results**

-Does the analysis presented match the analysis plan?

-Are the results clearly and completely presented?

-Are the figures (Tables, Images) of sufficient quality for clarity?

Reviewer #1: The results are not clearly and completely presented.

Figure 1 – prefer to present as tables with p-values

Table 1 

• Suggest to add 2 more columns of OPD (n = 113), IPD (n = 242) group in Table 1.

• Underlying diseases may be excluded because this contributes to more severe disease and complication? If underlying disease remained in the manuscript, please specify in each category of the nutritional status.

• Parasite infection, please specify. If only minor parasitic infection without fever or systemic signs and symptoms, they may be omitted in underlying diseases?

Table 2 & 3

• Suggest to add another 2 tables of % of DSS, DHF, DF and Severe, non-severe dengue in and please clarify category of severe dengue, whether shock, respiratory distress, bleeding or organ (s) failure.

• Separate OPD and IPD group will be presented as in Table 2 &3.

Results

• Page 5 – explanation of Figure 1, please add % of obese patients in both WHO 1997 and 2009 classifications. In WHO 1997, please add 5 of DSS and DHF separately. In WHO 2009, please add detail category of severe dengue, i.e. shock, respiratory distress, bleeding or organ (s) involvement.

• Page 7, first paragraph: please re-calculate % of normal weight (62.3%) and overweight/obese (28.7%). The number for normal weight = 164, overweight = 102 and overweight = 57 and the total number = 355 patients.

Reviewer #2: - For Figure 1, instead of the bar chart, the data should be represented in Table to more clarify and reduce repeated picture.

- The clinical characteristics of pediatric patients should be represented and shown the differences among groups.

Reviewer #3: Results

o Add an explanation of why cases were excluded (574 records were screened and 355 met inclusion/exclusion criteria) – could add information that XX were over 18y, xx did not have nutritional status, etc. 

• Table 2 and Table 3

o Incorrect labelling, the first column is not “n(%)” – that is rather for the 2nd and 3rd columns (and an accurate heading should be put for the first column information)

o Add what the reference group is (tables should be self-contained)

o Define all acronyms used in the table (tables should be self-contained)

**Conclusions**

-Are the conclusions supported by the data presented?

-Are the limitations of analysis clearly described?

-Do the authors discuss how these data can be helpful to advance our understanding of the topic under study?

-Is public health relevance addressed?

Reviewer #1: The conclusion will depend on the suggested study design.

Reviewer #2: The authors may discuss why to choose the BMI-for-age Z-score as an indicator based on WHO nutritional classification, instead of WHZ based on Thai Growth chart.

Reviewer #3: Discussion 

o It seems like the authors looked for something and did not find it.. but it is difficult as the reader to piece together why they were looking for it in the first place. From this paper, I was not convinced that this was worth looking for or what they really added to the literature on this topic. I would have expected an explanation of the biological plausibility, rationale, and implications for clinical work / policy that makes this topic worth examining. 

o A systematic review was cited so it seems this is worth examining but there was little to no justification of why this is helpful to look into for the reader of this paper. Add this so that the reader understands the biological plausibility and relevance, in particular, of an association between overweight/obesity and mild plasma leakage.

o Weight was taken on the first day of hospitalization. Add a point to the discussion of the potential bias that this may have (if weight loss already occurred during the disease progression prior to hospital admittance)? Or why this should *not* be an issue.

**Editorial and Data Presentation Modifications?**

Reviewer #1: As in the above suggestion

Reviewer #2: (No Response)

Reviewer #3: (No Response)

**Summary and General Comments**

Reviewer #1: To identify the association between nutritional status and dengue severity need the same population, i.e. only the admitted patients. OPD patients, about 1/3 of the study population in this study tend to have milder illness that may interfere with the results of the study.

Reviewer #2: Please specify how the data was extracted and by whom as well as reviewed and confirmed in the classifications.

Please clearly state inclusion/exclusion criteria with a brief rationale. 

Please reveal the clinical characteristics of patients.

Reviewer #3: What was the distribution of nutritional status of the general population that you were looking at (children in urban Thailand/Bangkok)? 

o To do this you could look at distribution of all at the clinic / or a comparison group (with another, clearly unrelated condition – or those children coming for wellness checks and therefore presumably healthy?)

o Published survey data from a large survey done in Bangkok recently (or at least in urban areas of Thailand?) 

o For example, this national survey has much lower %s of overweight and obesity: 

 � National survey of overweight/obesity: https://www.ncbi.nlm.nih.gov/pmc/articles/PMC5824639/

o This more recent article, however, mentions a survey with numbers more even higher than what was found in this study (36% obesity in high school students and 19.6% in primary-school-age children). But they don’t cite the specific survey.. (https://www.bumrungrad.com/en/health-blog/october-2019/child-obesity-becoming-a-growing-problem) 

o Overall, my point is that these numbers need some context. How is this population of children different from this age group living in Bangkok? Even without collecting/calculating anything further you can add this information. In particular, among those 11-17y as that makes up the vast majority of the sample.

Introduction

o The rationale why you are looking at this association in the first place remains unclear. Justify why it is thought that underweight and, in particular, overweight may be associated with more severe dengue outcomes.

o Be more explicit about what this will mean for clinicians and researchers – if you find there is an association.

o I would call them children and adolescents. It seems a bit misleading to say just children when almost 80% are 11-17 years.

PLOS authors have the option to publish the peer review history of their article (what does this mean?). If published, this will include your full peer review and any attached files.

Reviewer #1: Yes: Professor Siripen Kalayanarooj

Reviewer #2: No

Reviewer #3: No
---

## [Decision Letter · Decision Letter 1]

6 Apr 2022

Dear Dr. Limkittikul,

We are pleased to inform you that your manuscript 'Association between nutritional status and dengue severity in Thai children and adolescents' has been provisionally accepted for publication in PLOS Neglected Tropical Diseases.

Best regards,

Olaf Horstick, FFPH(UK)

Associate Editor

Sergio Recuenco

Deputy Editor

Reviewer's Responses to Questions

**Key Review Criteria Required for Acceptance?**

**Methods**

-Are the objectives of the study clearly articulated with a clear testable hypothesis stated?

-Is the study design appropriate to address the stated objectives?

-Is the population clearly described and appropriate for the hypothesis being tested?

-Is the sample size sufficient to ensure adequate power to address the hypothesis being tested?

-Were correct statistical analysis used to support conclusions?

-Are there concerns about ethical or regulatory requirements being met?

Reviewer #1: OK

**Results**

-Does the analysis presented match the analysis plan?

-Are the results clearly and completely presented?

-Are the figures (Tables, Images) of sufficient quality for clarity?

Reviewer #1: Yes.

**Conclusions**

-Are the conclusions supported by the data presented?

-Are the limitations of analysis clearly described?

-Do the authors discuss how these data can be helpful to advance our understanding of the topic under study?

-Is public health relevance addressed?

Reviewer #1: OK.

**Editorial and Data Presentation Modifications?**

Reviewer #1: In Author summary: second line: please modify the sentence ...that may kill infants and children.

Please add a sentence about comparing OPD and IPD cases in discussion.

**Summary and General Comments**

Reviewer #1: OK.

PLOS authors have the option to publish the peer review history of their article (what does this mean?). If published, this will include your full peer review and any attached files.

Reviewer #1: **Yes: **Professor Siripen Kalayanarooj

---

## [Editor Report · Acceptance letter]

13 May 2022

Dear Dr. Limkittikul,

We are delighted to inform you that your manuscript, "Association between nutritional status and dengue severity in Thai children and adolescents," has been formally accepted for publication in PLOS Neglected Tropical Diseases.

Best regards,

Shaden Kamhawi

co-Editor-in-Chief

Paul Brindley

co-Editor-in-Chief
